# Data augmentation for efficient learning from parametric experts

## Abstract

We present a simple, yet powerful data-augmentation technique to enable data-efficient learning from parametric experts. Whereas *behavioral cloning* refers to learning from samples of an expert, we focus here on what we refer to as the *policy cloning* setting which allows for offline queries of an expert or expert policy. This setting arises naturally in a number of problems, especially as a component of other algorithms. We achieve a very high level of data efficiency in transferring behavior from an expert to a student policy for high Degrees of Freedom (DoF) control problems using our *augmented policy cloning* (APC) approach, which combines conventional image-based data augmentation to build invariance to image perturbations with an expert-aware offline data augmentation approach that induces appropriate feedback-sensitivity in a region around expert trajectories. We show that our method increases data-efficiency of policy cloning, enabling transfer of complex high-DoF behaviors from just a few trajectories, and we also show benefits of our approach in the context of algorithms in which policy cloning is a constituent part.

## 1  Introduction

In various control and reinforcement learning settings, there is a need to transfer behavior from an expert policy to a student policy. Broadly, when only samples from the expert policy are available, the standard approach is to employ a version of regression from states to actions. This class of approaches for producing a policy is known as behavioral cloning [Pomerleau, 1989, Michie and Sammut, 1996]. Behavioral cloning is quite flexible and supports the setting where the expert trajectories come from a human teleoperating the relevant system directly, as well as various settings where the trajectories are sampled from other controllers, which themselves may have been trained or scripted. However, for any of the settings where the expert policy is actually available, rather than just samples from the expert, it is reasonable to suspect that sampling random rollouts from the expert policy followed by performing behavioral cloning is not the optimally efficient approach for transferring behavior from the expert to the student. Once a trajectory has been sampled via an expert rollout, there is actually additional information available that can be ascertained in the neighborhood of the trajectory, without having to perform an additional rollout, via the local feedback properties of the expert.

In this work, we refer to this setting, where we want to transfer from an expert policy to a student policy, while assuming the expert policy can be queried, as *policy cloning*. Naturally, there is still often an incentive to reduce the total number of rollouts, which may require actually collecting data in an unsafe or costly fashion, especially for real-world control problems. As such, there is an aim to characterize any efficiency that can be gained in learning from small numbers of rollouts without as much concern for how many offline queries are required of the expert policy. If one has primarily encountered behavioral cloning in the context of learning from human demonstrations, policy cloning,

with an available expert policy may seem contrived. However, policy cloning naturally arises in many settings. For example, we may have multiple experts that we wish to consolidate into a single neural network policy or there may be memory considerations that motivate compressing a large expert network into a smaller model with similar behavior. Perhaps most natural are settings in which an expert policy is costly or slow to execute, for example due to running a compute intensive procedure such as model predictive control (MPC) on specialized hardware (e.g. GPU); in such settings, the aim is to transfer expert behavior to a parametric student policy that amortizes the cost. DAGGER is one well known approach for efficiently transferring behavior from an expert to a student under these kinds of constraints [Ross et al., 2011]. In a separate setting, the expert may be suboptimal and the student needs to learn from expert while also being able to exceed the expert performance, perhaps by continuing to learn from a task via RL. This problem has been described as *kickstarting* in one incarnation [Schmitt et al., 2018], but also can arise when learning from behavioral priors [Tirumala et al., 2020], [Galashov et al., 2019], as also happens, for example, in Distral [Teh et al., 2017].

To improve data-efficiency in supervised settings generally, including in behavioral cloning settings, it is reasonable to consider data augmentation. Data augmentation refers to applying perturbations to a finite training dataset to effectively amplify its diversity, usually in the hopes of producing a model that is invariant to the class of perturbations performed. For example, in the well studied problem of object classification from single images, it is known that applying many kinds of perturbation should not affect the object label, so a model can be trained with many input perturbations all yielding the same output [Shorten and Khoshgoftaar, 2019]. This setting is fairly representative, with data augmentation usually intended to make the model "robust" to nuisance perturbations of the input. This class of image-perturbation has also been recently demonstrated to be effective in the context of control problems in the offline RL setting [Yarats et al., 2021, Laskin et al., 2020].

Critically, for control problems it is not the case that the action should be invariant to the input state. Or rather, while it does make sense for a control policy to be invariant to certain classes of sensor noise, an important class of robustness is that the policy is appropriately feedback-responsive. This is to say that for small perturbations of the state of the control system, the optimal action is different in precisely the way that the expert implicitly knows. This has been recognized and exploited in previous research that has distilled feedback-control plans into controllers [Mordatch and Todorov, 2014, Mordatch et al., 2015, Merel et al., 2019]. A similar intuition also underlies schemes which inject noise into the expert during rollouts to sample more comprehensively the space of how the expert recovers from perturbations [Laskey et al., 2017, Merel et al., 2019].

In this work, we leverage this insight to develop a highly efficient policy cloning approach that makes use of both classes of data augmentation. For a high-DoF control problem that operates only from state (humanoid run task from DeepMind control suite [Tunyasuvunakool et al., 2020]), we demonstrate the feasibility of policy cloning that employs state-based data augmentation with expert querying to transfer the feedback-sensitive behavior of the expert in a region around a small number of rollouts. Then on a more difficult high-DoF control problem that involves both state-derived and egocentric image observations (humanoid running through corrdiors task from DeepMind control suite [Tunyasuvunakool et al., 2020]), we combine the state-based expert-aware data augmentation with a separate image augmentation intended to induce invariance to image perturbations. Essentially our expert-aware data augmentation involves applying random perturbations to the state-derived observations, and training the student to match the expert-queried optimal action at each perturbed state, thereby gaining considerable knowledge from the expert without performing excessive rollouts simply to cover the state space around existing trajectories. Our approach compares favorably to sensible baselines, including the naive approach of attempting to perform behavioral cloning with state perturbations, which seeks to induce invariance (as proposed in [Laskin et al., 2020]) rather than feedback-sensitivity to state-derived observations.

In the presentation that follows, we will describe the problem setting (Section 2) as well as our approach (Section 3), describe the domains we employ and present our initial experiments (Section 4), show that our augmented policy cloning approach works well when used as a component of other algorithms like DAGGER and kickstarting (Section 5), and finally close with a discussion (Section 6).

## 2 Problem description

### 2.1 Expert-driven learning

We start by introducing a notion of expert-driven learning that will be used throughout the paper. At first, we present a general form of the expert-driven objective and then introduce a few concrete examples. We consider a standard Reinforcement Learning (RL) problem. We present the domain as an MDP with continuous states for simplicity, however the problem definition is similar for a POMDP with observations derived from the state. Formally, we describe the MDP in terms of a continuous state space $\mathcal{S} \in \mathcal{R}^n$ (for some $n > 0$), an action space $\mathcal{A}$, transition dynamics $p(s'|s, a) : \mathcal{S} \times \mathcal{A} \to p(\mathcal{S})$, and a reward function $r : \mathcal{S} \times \mathcal{A} \to \mathcal{R}$. Let $\Pi$ be a set of parametric policies, i.e. of mappings $\pi_\theta : \mathcal{S} \to p(\mathcal{A})$ from the state space $\mathcal{S}$ to the probability distributions over actions $\mathcal{A}$, where $\theta \in \mathcal{R}^m$ for some $m > 0$. For simplicity of the notation, we omit the parameter in front of the policy, i.e. $\pi = \pi_\theta$ and optimizing over the set of policies would be equivalent to the optimizing over a set of parameters. A reinforcement learning problem consists in finding such a policy $\pi$ that it maximizes the expected discounted future reward:

$$J(\pi) = \mathbb{E}_{p(\tau)} \left[ \sum_t \gamma^t r(a_t|s_t) \right], \tag{1}$$

where $p(\tau) = p(s_0) \prod_t p(a_t|s_t) p(s_{t+1}|s_t, a_t)$ is a trajectory distribution. We assume the existence of an expert policy $\pi_E(a|s)$. This policy could be used to simplify the learning of a new policy on the same problem. Formally, we construct a new learning objective which aims to maximize the expected reward of the problem in hand as well as to clone the expert policy:

$$J(\pi, \pi_E) = \alpha J(\pi) - \lambda D(\pi, \pi_E), \tag{2}$$

where $D$ is some measure of distance of $\pi$ from $\pi_E$ and $\alpha \geq 0, \lambda \geq 0$ are parameters measuring importance of both objectives. In most of the applications, $\alpha \in \{0, 1\}$ and $\lambda \geq 0$ represents a relative importance of cloning an expert policy with respect to the RL objective.

### 2.2 Behavioral cloning (BC)

One important instance of the objective (2) with $\alpha = 0$, $\lambda = 1$ is behavioral cloning. In this case, the measure of distance is defined as:

$$D_{BC}(\pi, \pi_E) = -\mathbb{E}_{(a,s) \in \mathcal{B}_E}[\log \pi(a|s)] \tag{3}$$

Here, $\mathcal{B}_E = \{(s_i, a_i), i = 1, \ldots, N\}$, $N > 0$ is a fixed dataset containing expert data. Minimizing the objective (3) is be equivalent to maximizing the likelihood of the expert data under the policy $\pi$. The action in eqn. (3) can be replaced by $\pi_E(s)$ for deterministic policies or by the mean or the mode for stochastic policies (e.g., by the mean $\mu_E(s)$ for Gaussian policies $\pi_E(\cdot|s) = \mathcal{N}(\mu_E(s), \sigma_E(s))$).

### 2.3 DAGGER

Performance of Behavioral Cloning (BC) can be limited due to the fixed dataset, since the resulting policy may fail to generalize to states outside the training distribution. A different approach, known in the literature as DAGGER [Ross et al., 2011] was proposed to overcome this limitation. In this setting, the expert is queried in states visited by the student, thus reducing distribution shift. In our notation, this corresponds to $\alpha = 0$, $\lambda = 1$ in eqn. (2) and the measure of distance is defined as:

$$D_{\text{DAGGER}}(\pi, \pi_E) = -\mathbb{E}_{p_\beta(\tau)}[\log \pi(a'_t|s_t)], \tag{4}$$

where $p_\beta(\tau)$, $\beta \in [0, 1]$ is a trajectory distribution where actions are sampled according to the mixture policy between a student and an expert:

$$p(a|s) = \beta \tilde{\pi}(a|s) + (1 - \beta)\pi_E(a|s), \tag{5}$$

The action $a'_t$ in eqn. (4) is resampled from the expert policy for the state $s_t$, i.e., $a'_t \sim \pi_E(\cdot|s_t)$. As in Section 2.2, for stochastic experts this action can be replaced by the mean or mode of the distribution in eqn. (4). The policy $\tilde{\pi}(a|s)$ corresponds to a frozen version of student policy $\pi$ so that the gradient $\nabla_\pi D_{\text{DAGGER}}(\pi, \pi_E)$ is not taken with respect to $p(a|s)$. Note that even though, in eqn. (4) we collect data from the environment, the setting nevertheless corresponds to pure imitation learning since expected reward is not directly maximized.

 **2.4 Kickstarting**

In eqn. (2), we combine both maximization of expected task reward and minimization of distance to the expert. In literature, it is known as *Kickstarting* [Schmitt et al., 2018]. In this case, in the objective from eqn. (2), $\alpha = 1$, and $\lambda \geq 0$. As the measure of distance, we use the cross-entropy from expert to a student, similarly to [Schmitt et al., 2018]:

$$J(\pi, \pi_E) = J(\pi) - \lambda \mathbb{E}_{p(\tau)} \left[ -\mathbb{E}_{\pi_E(a|s)} \log \pi(a|s) \right] \tag{6}$$

where $p(\tau)$ is a trajectory distribution, where actions are sampled according to the student policy $\pi(\cdot|s)$. Usually, in the *Kickstarting* setting, the expert is sub-optimal and the goal is to train a policy that eventually outperforms the expert. Thus, it is customary to reduce $\lambda$ over the course of training. Yet, for simplicity, in our experiments we keep this coefficient fixed.

# 3 Augmented policy cloning

The previous section has demonstrated that the objective corresponding to the cloning behavior from the parametric expert policy could arise in multiple scenarios. In this section we propose a new and simple method which can significantly improve the data efficiency of the approaches described in Section 2. We explain the basic idea for BC, but its generalization to other expert-driven learning approaches described in Section 2 is straightforward. In Section 5 we show results for these problems.

When optimizing the objective (3), for every state $s \in \mathcal{D}_E$ from the expert trajectories dataset, we consider a small Gaussian state perturbation:

$$\delta s \sim \mathcal{N}(0, \sigma_s^2) \tag{7}$$

which produces a new virtual state:

$$s' = s + \delta s \tag{8}$$

Then, for this state we query the expert and obtain a new action

$$a' \sim \pi_E(\cdot|s + \delta s) \tag{9}$$

We then augment the dataset $\mathcal{D}_E$ with these new pairs of virtual states and actions. More explicitly the idea can be expressed in terms of the following objective:

$$D(\pi, \pi_E)_{APC} = \mathbb{E}_{(a,s) \in \mathcal{B}_E} [\log \pi(a|s) + \mathbb{E}_{\delta s \sim \mathcal{N}(0, \sigma_s^2), a' \sim \pi_E(\cdot|s+\delta s)} \log \pi(a'|s + \delta s)] \tag{10}$$

We call this approach *Augmented Policy Cloning* (APC) as it queries the expert policy to augment the training data. This approach is different from a naive data-augmentation technique, where a new state would be generated, but associated with the original action (and not a new one). It therefore allows to build policies which are feedback-responsive with respect to the expert. We formulate APC algorithm for BC in Algorithm 1.

# 4 Core Results: Evaluation of Augmented Policy Cloning

## 4.1 Domains

To study how our method performs on complex control domains, we consider two complex, high-DoF continuous control tasks involving control of a physically simulated humanoid body. Both domains are implemented using the MuJoCo physics engine [Todorov et al., 2012] and are available in the `dm_control` repository [Tunyasuvunakool et al., 2020]. The first task is the standard control suite **Run** task, where the **Humanoid** body needs to run at a target speed and observations are based on proprioception. The second task is the **Walls** task which requires the same **Humanoid** body to run along a corridor and avoid walls, using both proprioception and egocentric vision as observations. Both of these problems are rather challenging insofar as they require stabilization and locomotion control of a relatively complex humanoid body with 21 actuated DoFs, in one case using vision to guide the movement. Note these environments are related to the domains that have been proposed for use in offline RL benchmarks [Gulcehre et al., 2020]; however, the experiments we perform in this work require availability of the expert policy, so we do not use offline data, but instead train new experts and perform experiments in the very low data regime. For more details, please refer to Section 1.1 in Supplementary Material.

---

**Algorithm 1** Augmented Policy Cloning (APC)

---

Parametric student policy: $\pi_\theta$
Initial parameters: $\theta_0$
expert policy: $\pi_E$
Dataset $\mathcal{B}_E = \{(s_i, a_i), i = 1, \ldots, N\}$, $N > 0$ of expert state-action pairs
State perturbation noise $\sigma_s$
Learning rate $\alpha$
Number of augmented samples: $M$
Number of gradient updates: $K$
Size of a batch: $L$
**for** k=1,...,K **do**
    Sample a batch of pairs $\{(a_i, s_i)\}_{i=1}^{L} \sim \mathcal{B}_E$
    For each state $s_i$, sample $M$ perturbations $\delta s_j \sim \mathcal{N}(0, \sigma_s), j = 1, \ldots, M$
    Construct $M$ virtual states $s'_{i,j} = s_i + \delta s_j, i = 1, \ldots, L, j = 1, \ldots, M$
    Resample new actions from expert $a'_{i,j} \sim \pi_E(\cdot|s'_{i,j})$
    For Gaussian experts, the action $a_i = \mu_E(s_i)$ and the new actions are $a'_{i,j} = \mu_E(s'_{i,j})$
    Compute the empirical negative log-likelihood:
$$\mathcal{L} = - \left[ \log \pi_{\theta_k}(a_i|s_i) + \frac{1}{M} \sum_{j=1}^{M} \log \pi_{\theta_k}(a'_{i,j}|s'_{i,j}) \right]$$
    Update the parameters $\theta_{k+1} = \theta_k - \alpha \nabla_\theta \mathcal{L}$
**end for**

---

For each task, we train expert policies to convergence using the MPO algorithm Abdolmaleki et al. [2018]. Since the expert policy essentially saturates task performance, for each task, we keep three partially trained experts such that we can assess the ability of the kickstarting approach to outperform sub-optimal experts. We refer to the different experts as **Low**, achieving approximatively 25 % of the optimal policy reward, **Medium**, achieving around 50 % of the performance and **High**, corresponding to the converged policy. Each expert is represented by a Gaussian policy. For more details, please refer to Section 1.2 in Supplementary Material.

## 4.2 Applying Augmented Policy Cloning

First, we evaluate the performance of APC in fitting a fixed dataset of expert trajectories. In order to study the data efficiency of the method, we construct datasets containing different numbers of expert trajectories. The expert policies are represented by conditionally Gaussian distributions, i.e. $\pi_E(\cdot|s) = \mathcal{N}(\mu_E(s), \sigma(s))$. Thus, to assess the robustness of our method to expert noise we produce trajectories using the experts' mean but adding different levels of (homoscedastic) zero-mean Gaussian noise $\sigma_E$.

$$a \sim \mathcal{N}(\mu_E(s), \sigma_E)$$

Note that in addition to policy noise $\sigma_E$ which is introduced when sampling trajectories, initial pose and environment layout (for the Walls task) are also sampled randomly for each episode. We consider 4 levels of expert policy noise: **Deterministic**, which uses the Gaussian mean for the action, **Low**, with $\sigma_E = 0.2$, **Medium** $\sigma_E = 0.5$ and **High** $\sigma_E = 1.0$.

For the APC method, we rely on Algorithm 1. For baselines, we consider BC algorithm from eqn. (3) as well as a simple modification of BC, where we apply, similar to APC, state perturbations as in eqn. (7) and eqn. (8), but we do not produce a new action from the expert. We call this approach Naive Augmented Behavior Cloning (Naive ABC) which essentially corresponds to robustification of the student policies with respect to state perturbation and is similar in spirit to standard data-augmentation approaches. For vision-based tasks, we consider random crop augmentations of size 48x48 (downsampled from the input image of 64x64), similar to Laskin et al. [2020]. When the image augmentations are used we add "with image" to the method name. On top of that, we consider a variant, where only image augmentation is used, which we call Naive ABC (image only). For all methods, as an action in the objective from eqn. (3), we use an expert mean $\mu_E(s)$. We train all approaches to convergence (300K learning iterations on Walls and 13M learning iterations on Run). Each learning iteration corresponds applying gradients to 64 trajectories, each containing 10 time steps. After each learning iteration, we evaluate the policy on both a validation set (50 random instances of the environment) and a test set (150 random instances of the environments).

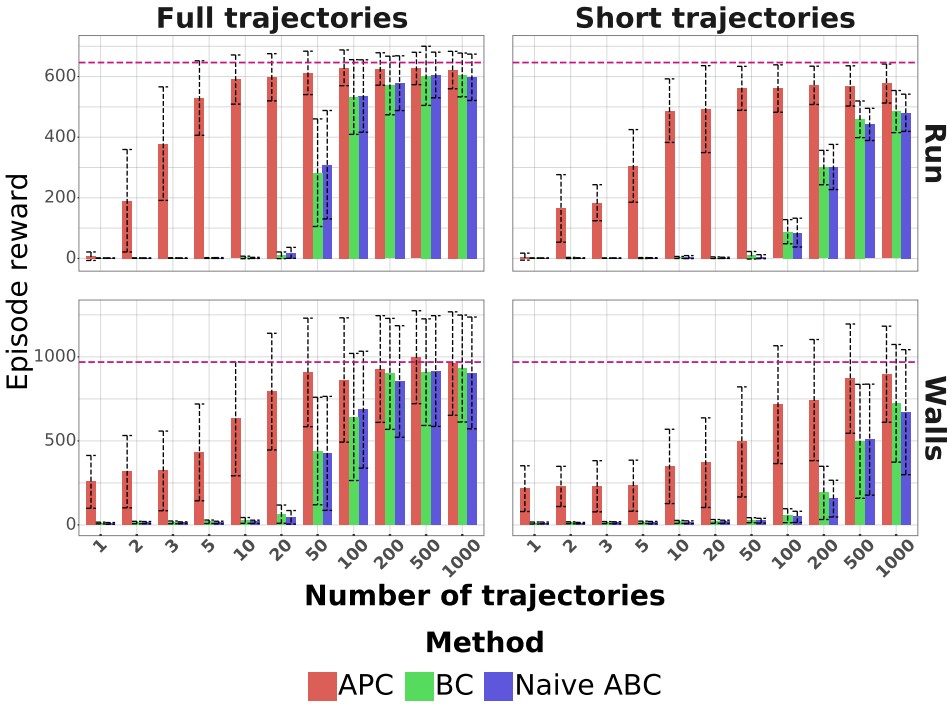

Figure 1: **Behavioral cloning results on Run and Walls tasks** (represented by rows). The X-axis represents the number of trajectories, whereas the Y-axis corresponds to the episodic reward averaged among 150 independent evaluations. The highest point of the bar corresponds to the mean, whereas the dashed lines indicate the standard deviation. The pink dashed line indicate average expert performance. The legend describes a method which is used. On the plot on the left depicts a standard BC experiment, where dataset contains a specified number of full trajectories from the expert. The plot on the right illustrates the experiment, where a dataset contains 1 full trajectories and the rest are the short ones, containing only 200 timesteps each.

We apply early-stopping based on the validation set performance to select the best model and report corresponding performance on the test set. For more details, please refer to Section 1.3 in Supplementary Material. As an additional evaluation, we test robustness of the obtained policies to a fixed amount of noise during execution. For a learned student policy $\pi(\cdot|s) = \mathcal{N}(\mu(s), \sigma(s))$, we evaluate it by executing an action:

$$a \sim \mathcal{N}(\mu(s), \sigma),$$

where $\sigma$ is the fixed amount of student noise. We consider similar noise magnitudes as for the expert. For APC and Naive ABC, we sweep over state perturbation noise levels and choose the ones performing the best on the validation set. For APC, we use $\sigma_s = 0.1$ for Run and $\sigma_s = 1.0$ for Walls. For Naive ABC, we use $\sigma_s = 0.001$ for Run and $\sigma_s = 0.01$ for Walls. The ablation experiments over noise levels for APC and Naive ABC are presented in Section 2 of the Supplementary Material (Figure 1 and Figure 2).

The first of results, in Figure 1 (left) demonstrates the increased data efficiency of APC over BC and Naive ABC in terms of number of trajectories. The noise level of the expert and student are fixed to **Low** for the ease of comparison. We also see that Naive ABC performs similarly to BC. To further push the limits of data efficiency, we conducted a variant where a dataset contains only 1 full trajectory (1000 timesteps for Run and around 2k timesteps for Walls) along with multiple short trajectories (200 time-step only). This dramatically reduces the amount of expert data available to learn from. However, we hypothesise that in the environments considered, much of the diversity of the trajectories arises due to initial state variation. This setting might arise in domains where execution is costly, such as robotics applications. In such setting we might have a few longer trajectories along with a patchwork of shorter trajectories covering more diverse parts of the state space. The results for this experiment are given in figure 1 (right). Again, we see that APC is significantly more efficient than BC and Naive ABC. Interesting to note that APC picks up quite a high performance after observing

10 (1 full and 9 short) trajectories for Run task and 100 (1 full and 99 short) trajectories for the Walls task. In Section 3 in Supplementary material, we provide additional results for Walls task when we use image-based perturbations.

In the next experiment, in order to understand how robust our method to noise, we study the impact of different levels of student and expert noises on performance. For each run, we use a dataset of 100 trajectories. The results are given in figure 2, where each column corresponds to a different level of expert noise, and the X-axis represents different levels of student noise. At first, we observe that APC is consistently more robust than BC and Naive ABC for any level of expert and student noise. On top of that, we can notice that for any fixed level of expert noise, the performance degrades when a student noise increases. Finally, we see that for higher noise levels of expert, the learned student performs better in the high noise regime. It is consistent with the intuition - training on noisy trajectories leads to a more robust policy. Overall, APC leads the most robust policy.

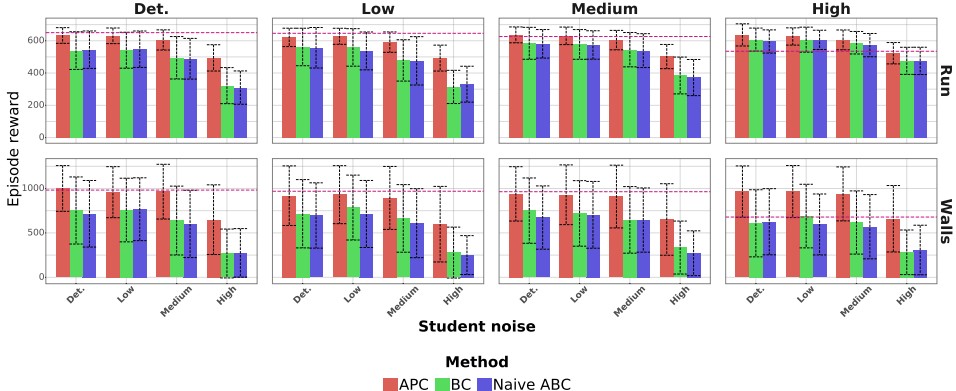

Figure 2: **Noise sensitivity results.** We consider 4 levels of noise for student and expert: **Deterministic**, which uses the Gaussian mean for the action, **Low**, is the noise $\sigma = 0.2$, **Medium** $\sigma = 0.5$ and **High** $\sigma = 1.0$. Each column corresponds to a different level of expert noise. X-axis corresponds to a different level of student noise. Y-axis corresponds to the episodic reward averaged among 150 independent evaluations. The highest point of the bar corresponds to the mean, whereas the dashed lines indicate the standard deviation. The legend denotes a method and a row corresponds to a task. The pink dashed line indicate average expert performance.

# 5 Additional Results: Augmented Policy Cloning as a subroutine

## 5.1 DAGGER with data augmentation

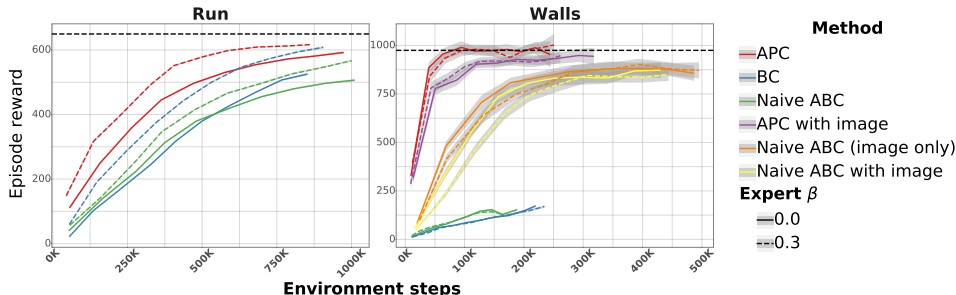

Figure 3: **DAGGER results**. On the X-axis we report the number of environment steps. On the Y-axis we report averaged across 3 seeds episodic reward achieved by the student. We report confidence intervals in the shaded areas. For Run task, the confidence intervals are very small and are not visible. In solid line we report the performance without using expert policy during the acting. In dashed line, we report the performance of the policy which mixes 30% with the expert. All the methods use mean action during evaluation.

As described in Section 2.3, DAGGER [Ross et al., 2011] is a more sophisticated approach where data is collected from the real environment by executing a policy from eqn. (5), which is a mixture between a student and an expert. In this section we study how data augmentation approaches affect the data efficiency of the DAGGER algorithm.

We consider similar baselines for both tasks as in the previous section. For an expert policy that has been pre-trained via MPO [Abdolmaleki et al., 2018], we perform online rollouts for two values of the expert-student mixing coefficient, $\beta = 0$ and $\beta = 0.3$ (see eqn. 5). Since both student and expert are Gaussian distributions, instead of using a $\log \pi$ in eqn. (4), we could use a state-conditional cross entropy from an expert to a student, $\mathcal{H}[\pi_E(\cdot|s)||\pi(\cdot|s)]$. Empirically, we found that it worked better than using $\log \pi$. We demonstrate a comparison in Section 4 in Supplementary Material. We run the experiments in a data-restricted setup such that for every collected trajectory (10 time-steps), we apply 10 gradient steps, using a replay-buffer to store the past experience. Additional experimental details are given in Section 1.4 in Supplementary Material. Results are shown in Figure 3. We see that APC and its vision variant outperform BC and Naive ABC similarly to the behavior cloning experiments. While we observe that image augmentation can help, we see that the primary advantage comes from the state-based augmentation for APC. For the Run task, we observe that all DAGGER methods achieve slightly lower performance than an expert policy. We speculate that this is due to insufficient coverage of the state space during training.

## 5.2 Kickstarting with data augmentation

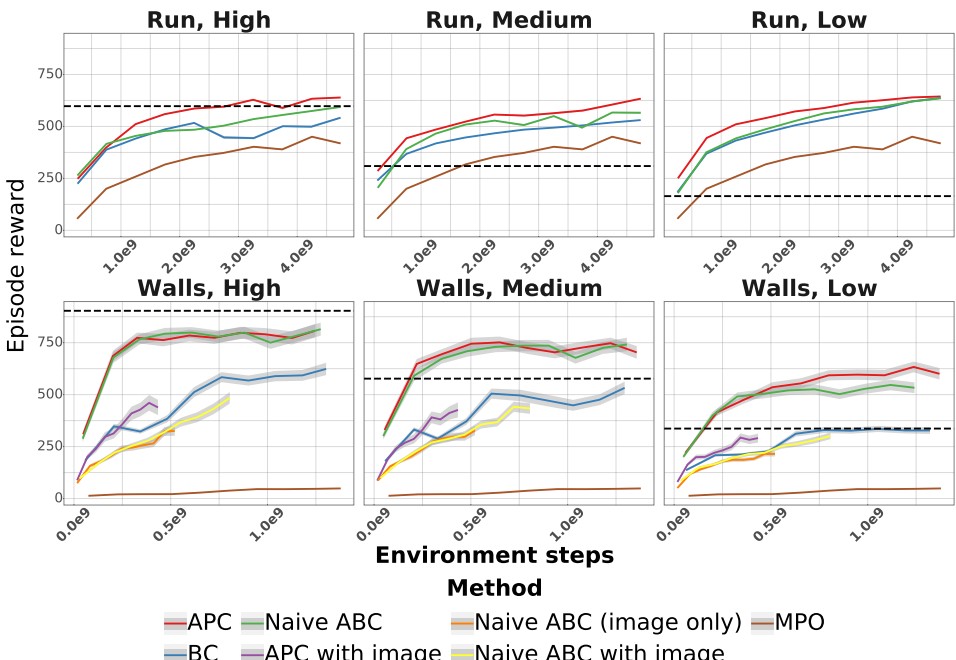

Figure 4: **Kickstarting results**. On the X-axis we show the number of environment steps. On the Y-axis we report averaged across 3 seeds episodic reward achieved by the student. We report confidence intervals in the shaded areas. For Run task, the confidence intervals are very small and are not visible. Each row indicates a task, whereas a column corresponds to the expert type. Dashed black line shows the expert performance.

A similar in spirit approach is kickstarting Schmitt et al. [2018], where we solve an RL task as well as cloning the expert policy. Similarly to previous section, we apply APC in kickstarting on the cross entropy term in eqn. (6). We use 3 types of expert policy as described in Section 4.1. We run experiments using a distributed setup with 64 acting policies and 1 learner, querying the batches of trajectories (of size 10) from a replay buffer. On top of running BC methods, we also report the performance of MPO Abdolmaleki et al. [2018] learning from scratch on the task of interest. All details are given in Section 1.5 in Supplementary Material. The results are given in figure 4.

We observe that APC performs better than Naive ABC on Run task and similarly on Walls task. Both approaches perform better than BC and learning from scratch. We hypothesise that the reason of not seeing a consistent advantage could be due to two factors. As we are in a high-data regime, since there is no limit on relative acting / learning ratio, and acting policies are not restricted to collect trajectories, it is unclear whether data-augmentation should help. In addition, we use reward signal which makes the impact of expert cloning less important. Note that the resulting agent is less data efficient in these experiments; this is because we do not control the relative ratio between acting and learning (i.e., no rate-limiting on the learner, due to instability of kickstarting experiments when rate-limiting was explored). Furthermore, unlike in kickstarting Schmitt et al. [2018], we do not use an annealing schedule of $\lambda$ to make the experiments simpler, but we still observe that a fixed coefficient helps to kickstart an experiment and outperform an expert policy. On top of that, we see that image-based augmentation have less of impact in this setting.

## 6 Discussion

Many expert-driven learning approaches actually have access to an expert that can be queried; however, this opportunity is rarely exploited fully. In this work we demonstrated a general scheme for more efficient transfer of expert behavior by augmenting expert trajectory data with virtual, perturbed states as well as the expert actions in these virtual states. This data augmentation technique is widely applicable and we demonstrated that it improves data efficiency when used in place of behavioral cloning both in the offline setting or when behavioral cloning is used as a step within DAGGER or kickstarting.

Critically, data efficiency is generally very important in realistic applications, where new data acquisition cost could be high. In particular, settings involving deployment of policies in the real world, such as robotics applications, may benefit from an ability to efficiently transfer expert policy behavior from one neural network to another (for compression or execution speed reasons), or to combine behavior from multiple experts into a single neural network. While overall, we consider the present work to be fairly basic research with limited ethical impact, insofar as our approach decreases the amount of data which needs to be collected through processes which could potentially be unsafe or costly, there is a potential positive social value.

The limitations of our approach consist in the reliance on the ability to query expert policy for the perturbed states which reduces the amount of applications where the method could be used. Another limitation is the reliance on the continuous state spaces. In discrete state spaces, it is unclear whether a small perturbation in state would result in a valid action from an expert.

In future work, we plan to explore how our proposed augmentation technique can be leveraged in the context of KL-regularized RL with behavior priors.

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
