# Data efficient learning from parametric experts: Supplementary material

## 1  Experimental details

### 1.1  Environment details

In this work we consider two environments from DeepMind Control suite [Tunyasuvunakool et al., 2020]: **Run** task with **Humanoid** body and Walls task (original name is "run through a corridor") with **Humanoid** body (original body was **CMU Humanoid**). In **Run** task, an agent must run at a target speed and gets reward which is proportional to the inverse distance between its current speed and the target speed. The agent receives proprioceptive observations only, which fully describe the state. In **Walls** task, an agent must run through the corridor and avoid the emerging walls. It receives reward which is proportional to the forward speed (through the corridor), thus incentivising it to run as fast as possible. It receives proprioceptive observations as well as the image of size 64x64 from the ego-centric camera. Action dimension is equal to 21. For more details, check Tunyasuvunakool et al. [2020].

### 1.2  Agent architecture

For all the experiments, we use the same agent architecture. The agent has two separate networks: actor (policy) and critic (Q-function). Both networks are split into 3 components: encoder, torso and head. Encoders for actor and critic are separate but have the same architecture. For state-only (no vision) observations, encoder corresponds to a simple concatenations of all the observations. For the vision input, it divides each pixel by 255 and then applies a 3-layer ResNet of sizes $(16, 32, 32)$ with $ELU$ activations followed by a linear layer of size 256 and $ELU$ activation. The resulting output is then concatenated together with state-based input. For actor network, torso corresponds to a 3 dimensional MLP, each hidden layer of size 256 with activation $ELU$ applied at the end of each hidden layer. The output of actor torso is then passed to the actor head network, which applies a linear layer (without activation) with output size equal to $N_a * 2$, where $N_a$ is the action dimension (21 in our case). It produces the actor mean $\mu$ and log-variance: $\log \tilde{\sigma}$. Then, the variance of the actor is calculated as

$$\sigma = softplus(\log \tilde{\sigma}) + \sigma_{min},$$

where $\sigma_{min} = 0.0001$. That would encode the Gaussian policy $\pi(\cdot|s) = \mathcal{N}(\mu(s)|\sigma(s))$. This parameterisation insures that the variance is never 0. The critic torso network is 1 dimensional MLP of size 256 with $ELU$ activation on top of it. Both critic encoder and critic torso are applied to the state input and not the action. The output of torso and the action are passed to the head, which firstly applies a $tanh$ activation to the action to scale it in $[-1, 1]$ interval, then concatenates both scaled action and torso output. This concatenated output is then passed through a 3-dimensional MLP with sizes $[256, 256, 1]$ with $ELU$ activations applied to all layers except the last one. This produces the Q-function representation, $Q(s, a)$. The critic network is not used for the Behavioral Cloning and DAGGER experiments.

To train experts and agents in kickstarting experiment, we use MPO [Abdolmaleki et al., 2018] algorithm. As for MPO-specific parameters, we used $\epsilon_\mu = 0.05$ and $\epsilon_\Sigma = 0.001$ as we found that using higher values for M-step constraints led to better kickstarting performance. For both tasks, we train MPO agents till convergence. In order to test the ability of the kickstarting method to

34th Conference on Neural Information Processing Systems (NeurIPS 2020), Vancouver, Canada.

outperform the sub-optimal experts, we keep partially trained expert policies by saving corresponding checkpoints. Then, we keep 3 types of experts: **Low**, the one achieving approximately 25% of the converged expert performance, **Medium**, the one achieving around 50 % of expert performance and **High**, the one achieving the expert performance.

## 1.3 Behavioral cloning experiment details

For behavioral cloning experiments, for both tasks, we consider converged experts (expert type is **High**). Given this expert, we produce fixed size datasets (for different number of trajectories), by unrolling the expert with a fixed noise level. Since the expert is a Gaussian policy, $\mathcal{N}(\mu_E(s), \sigma_E(s))$, we unroll the trajectories with actions sampled with a fixed noise level $\sigma$:

$$a_E \sim \mathcal{N}(\mu_E(s), \sigma)$$

We consider 4 different levels of $\sigma$: **Deterministic**, meaning that we unroll the expert trajectories using only the mean $\mu_E$, **Low**: $\sigma = 0.2$, **Medium**: $\sigma = 0.5$, **High**: $\sigma = 1.0$. We also tried values in-between, but did not found a qualitative difference. We also tried values above $\sigma = 1.0$, but the performance for these ones was almost zero. We unroll the expert trajectories by chunks containing 10 time steps each and put it in a dataset. We use Reverb (from ACME [Hoffman et al., 2020]) backend for this. A full trajectory for a Run task corresponds to 1000 time steps which corresponds to 25 seconds of control time with a control discretization of 0.025 seconds. A full trajectory for the Walls task corresponds to 2000-2500 time steps. This variation is due to potential early stopping of the task execution (in case if the agent falls down). The discretization for the control is 0.03 seconds and maximum episode length is 45 seconds.

For each of the task and the dataset variant (number of trajectories or/and expert noise level), we train a policy via Behavioral cloning by optimizing the objective from eqn. (3) from the main paper, where the expert action is replaced by the expert mean $\mu_E(s)$. More formally, we apply the Algorithm (1) from the main paper where for BC and naive-ABC we either do not apply state perturbation or do not produce a new expert action. For all the experiments we use a learning rate $\alpha = 0.0001$, number of augmented samples $M = 10$, batch size of $L = 64$. We run the experiment for $K = 13M$ steps for Run task and for $K = 300K$ steps for Walls task. For each learning iteration, we evaluate the model on 50 random instances of environment, which corresponds to a validation set, and on 150 random instances of environment corresponding to a test set. We use validation set to select parameters, such as state perturbation noise $\sigma_s$ as well as to select among the converged models (as we observed that when the models converged, there were always small variations in performance). We then reported the performance on the test set for all the plots. The values of state perturbation noise for APC are: $\sigma_s = 0.1$ for Run and $\sigma_s = 1.0$ for Walls task. For Naive ABC, these values are: $\sigma_s = 0.001$ for Run and $\sigma_s = 0.01$ for Walls tasks. The values which we tried are: $[0.0001, 0.001, 0.01, 0.05, 0.1, 1.0, 2.0, 10.0]$. For the experiments using short trajectories, the datasets were built by adding 1 full trajectory (1000 time steps for Run task and around 2000-2500 time steps for Walls tasks) with the rest containing only short trajectories (fixed to 200 time steps).

## 1.4 DAGGER experiment details

For DAGGER, similarly to BC experiments, we use **High** type of the expert (converged one). Throughout the experiment, we use the replay buffer of size $1e6$ where each element corresponds to 10-step trajectory, implemented using Reverb (from ACME [Hoffman et al., 2020]). We use the actor-learning architecture, with 1 actor and 1 learner, where the actor focuses on unrolling current policy and on collecting the data, whereas the learner samples the trajectories from the replay buffer and applies gradient updates on the parameters. When doing so, we control a relative rate of acting / learning via rate limiters as described in Hoffman et al. [2020] such that for each time step in the trajectory, we apply in average 10 gradient updates. This allows us to be very data efficient and get the full power from the data augmentation technique. In order to achieve it, we set the samples per request (SPI) parameter of the rate limiter to be $T * B * 10$, where $T = 10$ is the trajectory length (sample from a replay buffer), $B$ is the batch size (256 for Run and 32 for Walls). When sampling from the replay buffer, we use uniform sampling strategy. When the replay buffer is full, the old data is removed using FIFO-strategy.

For each method and each domain, we run the experiment with 3 random seeds. Normally, in DAGGER, the parameter $\beta$ of mixing the experience between the student and an expert, should

decrease to 0 throughout the learning. For simplicity of experimentation, we used fixed values. We report the results using $\beta = 0$ and $\beta = 0.3$, but we also experimented with values $\beta = 0.1, 0.2, 0.4, 0.5$. We found that our chosen values provided most of the qualitative information. The values of state perturbation noise for APC are: $\sigma_s = 0.1$ for Run and $\sigma_s = 1.0$ for Walls task. For Naive ABC, these values are: $\sigma_s = 0.01$ for Run and $\sigma_s = 0.001$ for Walls tasks. The values which we tried are: $[0.00001, 0.0001, 0.001, 0.01, 0.1, 1.0, 10.0]$.

When collecting the data, we use a mixture of student and expert, which are represented as stochastic policies via Gaussian distributions. For evaluation, we used their deterministic versions, by unrolling only the mean actions.

To train policies via DAGGER, we used analytical cross-entropy between expert and student instead of log probability of student on expert mean actions, as we found that it worked better in practice. We provide qualitative comparison in Section 4 in Supplementary material.

## 1.5 Kickstarting experiment details

For kickstarting experiment, we use 3 types of expert policy, as described in Section 1.2 in Supplementary material. We run experiments using a distributed setup with 64 actors and 1 learner, which queries the batches of trajectories (each containing 10 time steps) from a replay buffer of size $1e6$. We use Reverb (from ACME [Hoffman et al., 2020] as a backend. Batch size is 256 for Run and 32 for Walls. We run the sweep over $\lambda$ parameter from eqn. (6) from the main paper. As opposed to DAGGER, we do not use the rate-limiter to control the relative ratio between acting and learning as we found that kickstarting in such a regime was unstable. We found that $\lambda = 0.0001$ worked best for Run, whereas $\lambda = 0.01$ worked best for Walls. The values we tried are: $[0.0001, 0.001, 0.01, 0.1, 1.0, 10.0]$. We found that for higher values of $\lambda$, the learning was faster but the resulting policy did not outperform the expert. On top of running BC methods, we also report the performance of MPO Abdolmaleki et al. [2018] learning from scratch on the task of interest. The values of state perturbation noise for APC are: $\sigma_s = 0.01$ for Run and $\sigma_s = 0.01$ for Walls task. For Naive ABC, these values are: $\sigma_s = 0.00001$ for Run and $\sigma_s = 0.0001$ for Walls tasks. The values which we tried are: $[0.00001, 0.0001, 0.001, 0.01, 0.1, 1.0, 10.0]$.

## 1.6 Computational resources

For pretraining two expert policies on two tasks, we used in **total** 1 **GPU v100 and 64 4-core CPUs and around 500Gb memory for each run.**

For Behavioral cloning experiments, we used 1 GPU p100 and 1 CPU with 4 cores and around 64GB of memory for each run. In total we had a sweep over state-noise perturbation for both APC and Naive ABC, where sweep contained 8 experiments, resulting in 16 experiments in total (it would correspond to Figure 1 and Figure 2 from the supplementary material), which should be multiplied by 2 as we use 2 tasks, resulting in **32 experiments each with 1 p100 GPU, 1 CPU with 4 cores and around 64GB of memory**.

To produce Figure 1 in the main paper, we run 11 (different number of trajectories) * 3 (number of methods) * 2 (full or short trajectories) = 66 experiments. To produce Figure 2 in the main paper, we run 4 (different expert noises) * 3 (number of methods) = 12 experiments. To produce Figure 3 from supplementary material, we ran additionally 6 (different number of trajectories) * 3 (additional methods) * 2 (short or full trajectories) = 36 experiments. On top of that, we need to multiply it by 2 (as we have 2 tasks), resulting in 114 * 2 = 228 experiments. **Therefore, for behaviour cloning part, we ran in total 260 experiments, each using 1 p100 GPU and 1 CPU with 4 cores and around 64Gb of memory**

For each DAGGER experiment, we use 1 v100 GPU, 4 CPU with 4 cores each (for Replay, actor, evaluator, and remover), around 100Gb of memory in total. We conduct 3 (different methods) * 3 (different seeds) * 2 (different beta) * 2 (different tasks) = 36 experiments for non-vision based methods and 3 (different methods) * 3 (different seeds) * 2 (different beta) = 18 experiments for vision based methods. On top of that, we conducted a sweep over $\sigma_s$ for APC and Naive ABC and for each task, which leads 84 other experiments. **In total, we conducted 138 experiments each using v100 GPU, 4 CPU with 4 cores and 100Gb memory.**

For each kickstarting, we used 1 v100 GPU for walls and 1 p100 GPU for run tasks, 74 CPU with 4 cores each (64 for actors, 8 for evaluators, 1 for replay, 1 for remover), around 800Gb of memory in total. We conducted 84 experiments to sweep over $\sigma_s$ for APC and Naive ABC, and 36 experiments to sweep over $\lambda$ parameter. On top of that, we conducted in total 3 (different methods) * 3 (different seeds) * 3 (different experts) * 2 (different tasks) = 54 experiments for non-vision based methods and 3 (different methods) * 3 (different seeds) * 3 (different experts) = 27 experiments for vision-based methods. **In total, it would correspond to 200 experiments, each half of which used v100 GPUs and half of which used p100 GPU, 74 CPU with 4 cores and around 800Gb of memory.**

### 1.7 Plotting details

When we plot the results in Figure 3, Figure 4 from the main paper and in Figure **??** and Figure 5 in Supplementary material, we use the following method. For each independent task, method and independent run (seed), we split the data into bins, each containing 10% of the data. Then, in each bin, the performance is averaged as well as the 95% confidence interval is calculated. We then report these values in the figure.

## 2  APC and ABC state noise ablations

In this section we provide additional ablations for the state-noise perturbation level $\sigma_s$ from the eqn. (7) from the main paper. In Figure 1, we show the results for APC, whereas in Figure 2, we show the results for Naive ABC. We see that there is a sweet spot for the state perturbation noise level.

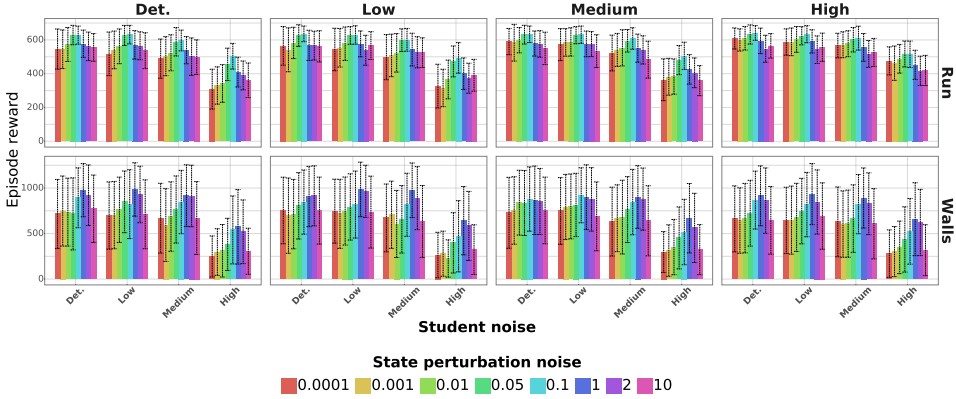

Figure 1: **State perturbation noise sensitivity for APC.** In this plot we represent the APC method trained on 100 full trajectories sampled under different level of expert noise which is represented by different columns. On the X-axis is the different level of a student noise at evaluation time. The legend denotes different levels of a state perturbation noise $\sigma_s$ from the eqn. (7) from the main paper. The Y-axis represents the averaged among 150 independent evaluations episode reward.

## 3  Additional comparisons for Walls task

In Figure 3, we provide additional results for behavioral cloning experiment on Walls task where we try different variants of APC and Naive ABC with additional image-based augmentation as described in the main paper.

## 4  Objective functions comparison for DAGGER

In Figure 4 and in Figure 5, we provide ablations over different objectives for DAGGER with $\beta = 0.0$ and $\beta = 0.3$ correspondingly. We see that overall, training with cross-entropy leads to better results than with log prob on the mean action, especially when $\beta = 0.0$.

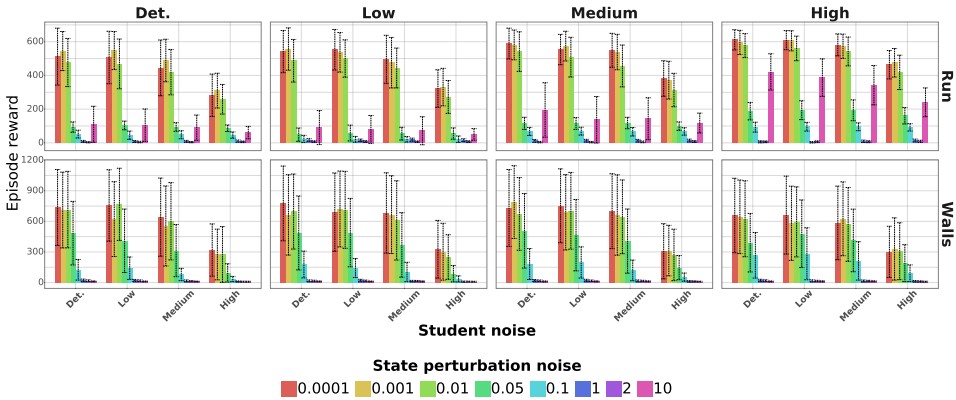

Figure 2: **State perturbation noise sensitivity for ABC**. In this plot we represent the APC method trained on 100 full trajectories sampled under different level of expert noise which is represented by different columns. On the X-axis is the different level of a student noise at evaluation time. The legend denotes different levels of a state perturbation noise $\sigma_s$ from the eqn. (7) from the main paper. The Y-axis represents the averaged among 150 independent evaluations episode reward.

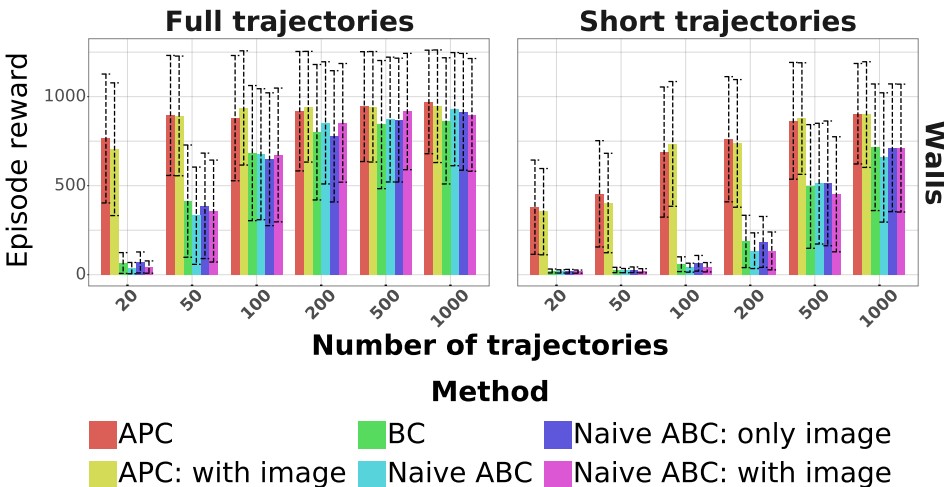

Figure 3: **Additional behavioral cloning results on Walls tasks with additional methods added.** X-axis corresponds to a number of trajectories used in each of the dataset. The Y-axis corresponds to the episodic reward. The high point of the bar plot corresponds to the average value (averaged across 150 independent evaluations) and the dashed lines indicate the standard deviation. The pink dashed line indicate average (among the same 150 independent evaluations) expert performance. The legend describes a method which is used. On the plot on the left depicts a standard BC experiment, where dataset contains a specified number of full trajectories from the expert. The plot on the right illustrates the experiment, where a dataset contains 1 full trajectories and the rest are the short ones, containing only 200 timesteps each.

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

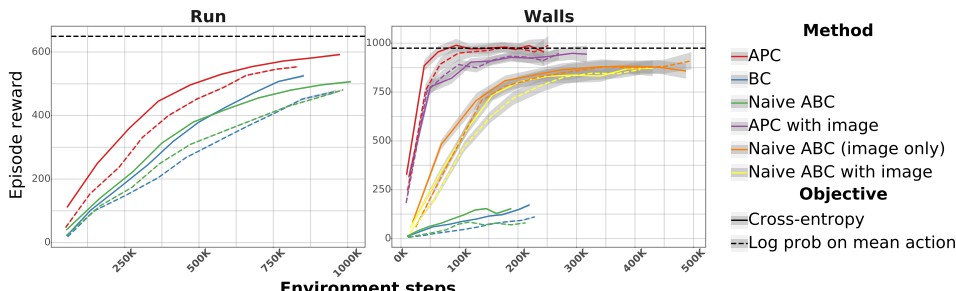

Figure 4: **DAGGER objective sweep with** $\beta = 0.0$. On the X-axis we report the number of environment steps. On the Y-axis we report averaged across 3 seeds episodic reward achieved by the student. Shaded area corresponds to confidence intervals. For a Run task, the confidence intervals are small, so they are not visible. In solid line we report the performance when training using the cross-entropy. In dashed line, we report the performance when training using log probability on the mean action from the expert. All the methods use mean action during evaluation. The black dashed line indicate average (among the same 150 independent evaluations) expert performance for the given expert noise level.

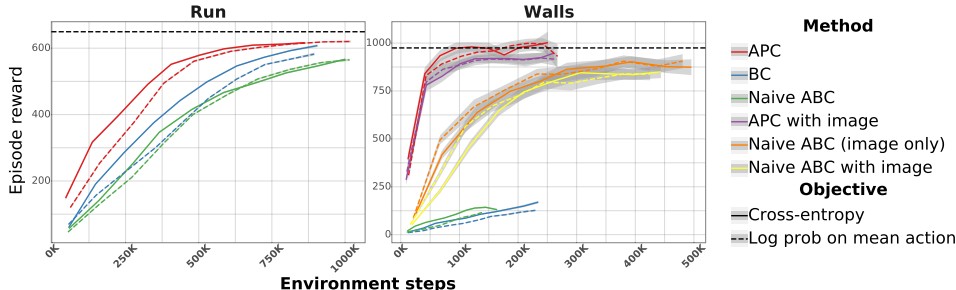

Figure 5: **DAGGER objective sweep with** $\beta = 0.3$. On the X-axis we report the number of environment steps. On the Y-axis we report averaged across 3 seeds episodic reward achieved by the student. Shaded area corresponds to confidence intervals. For a Run task, the confidence intervals are small, so they are not visible. In solid line we report the performance when training using the cross-entropy. In dashed line, we report the performance when training using log probability on the mean action from the expert. All the methods use mean action during evaluation. The black dashed line indicate average (among the same 150 independent evaluations) expert performance for the given expert noise level.

Saran Tunyasuvunakool, Alistair Muldal, Yotam Doron, Siqi Liu, Steven Bohez, Josh Merel, Tom Erez, Timothy Lillicrap, Nicolas Heess, and Yuval Tassa. dm_control: Software and tasks for continuous control. *Software Impacts*, 6:100022, 2020.