# OpenReview forum: "Data augmentation for efficient learning from parametric experts"
_NeurIPS.cc/2021/Conference — NeurIPS 2021 Submitted_

### Official Review · Reviewer_u3Dw · 2021-07-13

**Rating:** 3
**Confidence:** 5

**Summary:**

This paper proposes to study the non-interactive imitation learning setting where access to the parametric expert policy is available. The paper proposes to augment the logged expert data’s states with noise, and then determine then sample corresponding actions from the expert’s policy. These perturbed states and corresponding action are then use to train the agent. They show this method outperforms Behavior cloning and an approach where the logged states are perturbed but the corresponding actions are kept fixed on imitation learning tasks and RL tasks where IL is used for warmstarting.

**Limitations And Societal Impact:**

The setting is contrived and the proposed method seems obvious given the setting.

**Main Review:**

The paper is well-written and clear, but the setting seems fairly contrived. The examples they give where this problem may arise are not reflected in their experimental setup. For instance, they say the setting may come up when there are multiple experts which must be combined into one (this is not totally clear either, what do you do when the experts disagree? Do you randomly sample?), or if the expert is an expensive MPC procedure. However, in their experiments the experts have the same network architecture as the agent. In the examples they describe, there may be other issues, for example non-realizability which could complicate things.

In any case, given the setting, the proposed method seems more like an obvious first baseline to try rather than an original contribution. It’s completely expected that it would perform better than BC and the other naive method, so there isn’t much insight or takeaway from the experiments.

**Time Spent Reviewing:**

1.5

---

> ### Author Response · Authors · 2021-08-10
> **Response to u3Dw**
>
> **We thank u3Dw for taking time to review our work. Please find below our response which addresses your concerns.**
>
> >*This paper proposes to study the non-interactive imitation learning setting where access to the parametric expert policy is available.*
>
> **While this is a mostly fair characterization, please note that the proposed method is not limited to non-interactive imitation learning. Our approach is also compatible with the online setting, as is supported by our experiments with DAgger and kickstarting. Furthermore, please note that the method does not essentially rely on the parametric policy, but on the ability to query it for any state.**
>
> >*The examples they give where this problem may arise are not reflected in their experimental setup. For instance, they say the setting may come up when there are multiple experts which must be combined into one (this is not totally clear either, what do you do when the experts disagree? Do you randomly sample?), or if the expert is an expensive MPC procedure. However, in their experiments the experts have the same network architecture as the agent. In the examples they describe, there may be other issues, for example non-realizability which could complicate things.*
>
> **There are many motivating examples for this method in different contexts. It is more than it would be reasonable for us to access in a single paper. We chose to evaluate the method in following algorithmic settings - simple Behavioral Cloning, DAgger and kickstarting, but we wanted to make clear that there are other settings outside of these as well. We will clarify this in the paper.**
>
> >*In any case, given the setting, the proposed method seems more like an obvious first baseline to try rather than an original contribution. It’s completely expected that it would perform better than BC and the other naive method, so there isn’t much insight or takeaway from the experiments.*
>
> **The proposed method may seem to you obvious in hindsight. However, we assert that it is a quite different approach than what is typically used for data augmentation. As described in the paper, the typical approach for data augmentation in supervised learning consists in perturbing inputs but producing the same output, thus encouraging the robustness of the model. In RL, such techniques are used in [1], including Gaussian state perturbation, which in our paper is referred to as Naive ABC. Our method, however, is different, because on top of perturbing inputs, we also produce different outputs (novel actions from experts given perturbed state), encouraging the model to be feedback-responsive rather than invariant. While we point to some approaches which perform related techniques [2-4], we are unaware of any papers which systematically present an approach such as ours, which raises a question about whether this method is that obvious. One of your concerns could be that this is an obvious solution to a problem that is not frequently encountered.  However, we do not believe that this setting is actually uncommon. As we presented in the paper, the ability of querying an expert policy with continuous states may rise in different Behavior Cloning situations as well as in such widely used approaches as DAgger and kickstarting. In addition, what is more striking about the method is how much benefit it brings - see for example, Figure 1 on how much more data efficient BC becomes or Figure 3 on how it improves DAgger algorithm. Even if it may be expected that such a technique would help, we demonstrate that it actually helps quite a lot - which is solid evidence of why one wants to use it. If it was the case that it just marginally improved performance, then even if the technique may have seemed obvious, it would have been hard to argue for its usage.**
>
> References:
>
> [1] Reinforcement learning with augmented data, Laskin et al., 2020
>
> [2] Neural probabilistic motor primitives for humanoid control, Merel J. et al., 2019
>
> [3] Combining the benefits of function approximation and trajectory optimization, Mordatch I. et al., 2014
>
> [4] Interactive control of diverse complex characters with neural networks, Mordatch I. et al., 2015

---

### Official Review · Reviewer_HWtk · 2021-07-14

**Rating:** 3
**Confidence:** 4

**Summary:**

The paper introduces an approach to learning from an expert where a learning policy can query the expert. In particular, the authors propose to perturb the original states with Gaussian noise and then query expert actions for these states.

**Limitations And Societal Impact:**

Yes.

**Main Review:**

### Novelty
The paper has limited novelty. The idea of augmenting the states with Gaussian noise is known to improve the robustness of reinforcement learning on mujoco tasks (see [1, 2]). While learning from an oracle is studied in DAgger.

### Strengths
The paper demonstrates improvements over several baselines including behavioral cloning and image-based behavioral cloning with standard image augmentations.

### Weaknesses
I believe that the major weakness of this work is that it makes a strong assumption that adding Gaussian noise will produce a correct state. This assumption might not hold on to a variety of tasks. Moreover, the approach is evaluated on a limited set of tasks that raises further concerns regarding the generality of the approach.

I cannot recommend the paper for acceptance in its current form. In particular, I have concerns regarding the generality of the approach. The experimental evaluation is limited.

[1] Reinforcement learning with augmented data, Laskin etal., 2020

[2] S4RL: Surprisingly Simple Self-Supervision for Offline Reinforcement Learning, Sinha, etal., 2021

**Time Spent Reviewing:**

5

---

> ### Author Response · Authors · 2021-08-10
> **Response to HWtk**
>
> **Thank you HWtk for taking time to review our paper. Please find our response below addressing your concerns.**
>
> >*The paper has limited novelty. The idea of augmenting the states with Gaussian noise is known to improve the robustness of reinforcement learning on mujoco tasks (see [1, 2]). While learning from an oracle is studied in DAgger.*
>
> **While the idea of augmenting states with Gaussian noise is known to improve the robustness of RL tasks, as you pointed out in references, in this work we do not seek to improve robustness/invariance, but rather to improve data-efficiency. Our method differs from simply augmenting states with Gaussian noise, because we also produce an appropriate new action for the perturbed state. In fact, we are seeking a policy which is feedback responsive with respect to the perturbed state from the expert policy. It differs from the methods proposed in references and from classical data augmentation techniques, which indeed, seek robustification / invariance.**
>
> >*I believe that the major weakness of this work is that it makes a strong assumption that adding Gaussian noise will produce a correct state. This assumption might not hold on to a variety of tasks.*
>
> **You correctly observe that a perturbed state may no longer be a valid state.  However we do not believe this is an issue.  Essentially,  training a student on invalid states will simply result in a student that has no guarantees about producing sensible behavior on the invalid states that it will never encounter (and indeed cannot encounter at all) when it is deployed.  The more relevant concern is whether the student knows what action to produce in states that it can encounter, and empirically, we demonstrate that training on perturbed states does fairly consistently produce student policies that act effectively for diverse states.**
>
> >*Moreover, the approach is evaluated on a limited set of tasks that raises further concerns regarding the generality of the approach.*
>
> **In fact, before presenting the results, we tried our method on a few other simpler DM control suite environments - Humanoid Walk, Walker Walk and Run tasks, and we saw similar results. We decided to focus our presentation on Humanoid Run and Humanoid Walls tasks because these lie on a hard part of the spectrum of the DM control suite and are generally quite challenging to solve. Results on other simpler tasks were very similar and did not add much value, so we decided not to include them. For the camera-ready version, we will add results on more tasks.**

---

### Official Review · Reviewer_sn3m · 2021-07-15

**Rating:** 6
**Confidence:** 3

**Summary:**

This paper proposed an data-augmentation technique -- "augmented policy cloning (APC)" to enable efficient policy learning from parametric experts. It achieve a high level efficiency in transferring knowledge from an expert to a student policy for high Degrees of Freedom environment. The augmentation is introduce not only for the states but also to the actions of underlying policy.

**Limitations And Societal Impact:**

Yes

**Main Review:**

General:
The paper is well-written and easy to follow even for one who is not quite familiar with the area.
The idea is simple, yet very interesting. I am not quit into related literature, so I would not comment over its novelty.
My greatest concern is that whether this setting- "cloning a parameterised experts" is a large restriction.
Experiments are decent. However, there is space to improve.

Major:
- More discussion on the real world use case of the proposed method. To me, a prameterized expert is a strong restriction that would hinder the use case of proposed method.

- The main results Fig 1-4 are presented over two environments (Runs & Walls). I would like to see how this method perform over other environments.

- A followup to last one would be to analysis the noise scale used in the method when running on other enviroments. In other words, I would like the author to further analysis what aspect of the environment or expert policy would affect the noise scale.

Minor:

- line103: No \pi in J(\pi)'s expression. p(a|s) --> \pi(a|s)
- the presentation of Figure 2 might be improved

**Time Spent Reviewing:**

5-8

---

> ### Author Response · Authors · 2021-08-10
> **Response to sn3m**
>
> **Thank you sn3m for taking time to review our work. Please find below our response addressing your concerns.**
>
> >*My greatest concern is whether this setting- "cloning a parameterised experts" is a large restriction.*
>
> **We would like to clarify that our method can be applied in any setting where the expert can be queried (i.e., it does not need to be of a known parametric form, but could also be a black-box function). Methods do not apply everywhere. We do describe the regime that this works for, which is not universal, but is a very relevant regime. Other methods, such as DAgger, are also restricted in their domain of applications (in fact DAgger and kickstarting assume a similar ability to query an expert).**
>
> >*Experiments are decent. However, there is space to improve.*
>
> **Thank you, we plan to add more environments in the camera-ready version as well as to improve the clarity of the experimental section.**
>
> >*More discussion on the real world use case of the proposed method. To me, a prameterized expert is a strong restriction that would hinder the use case of proposed method.*
>
> **As discussed above, the parameterised expert or queryable expert assumption is not unreasonable in a number of applications. Both DAgger (widely used technique) and kickstarting assume a similar ability to query an expert. Behavior Cloning (BC) could also be done when experts could be queried. As discussed in the Introduction Section of the paper, such a setting could arise in multiple scenarios. For example, we may have multiple experts that we wish to consolidate into a single neural network policy or there may be memory considerations that motivate compressing a large expert network into a smaller model with similar behavior. Other setting would entail a situation where expert policy is costly or slow to execute, for example due to running a compute intensive procedure such as model predictive control (MPC) on specialized hardware (e.g. GPU). Finally, the expert could be trained using different architecture, different sets of observations (with potentially additional privileged information to simply training) as well as on a different but related task. We have attempted to communicate these various situations in the main text and demonstrate that an ability to query an expert (or having a parameterized expert) is not a strong restriction. We will re-emphasize this point.**
>
> >*The main results Fig 1-4 are presented over two environments (Runs & Walls). I would like to see how this method perform over other environments.*
>
> **In fact, before presenting the results, we tried our method on a few other simpler DM control suite environments - Humanoid Walk, Walker Walk and Run tasks, and we saw similar results. We decided to focus our presentation on Humanoid Run and Humanoid Walls tasks because these lie on a hard part of the spectrum of the DM control suite and are generally quite challenging to solve. Results on other simpler tasks were very similar and did not add much value, so we decided not to include them. For the camera-ready version, we will add results on more tasks.**
>
> >*A followup to last one would be to analysis the noise scale used in the method when running on other enviroments. In other words, I would like the author to further analysis what aspect of the environment or expert policy would affect the noise scale.*
>
> **We provide ablations over the noise scale in Figure 1 and Figure 2 in Supplementary material.**
>
> >*line103: No \pi in J(\pi)'s expression. p(a|s) --> \pi(a|s)*
>
> **Thank you, we will fix it in the camera ready version.**
>
> >*the presentation of Figure 2 might be improved*
>
> **Thank you, we will work on improving Figure 2 for the camera-ready version**

---

### Official Review · Reviewer_KLas · 2021-07-15

**Rating:** 4
**Confidence:** 4

**Summary:**

The paper proposes a policy cloning method that transfers an expert to a student. In the considered setting, the expert policy is accessible at all times, while there is no access to the environment to perform RL. The proposed method leverages data augmentation to successfully train the student policy.

**Limitations And Societal Impact:**

Please see the main review.

**Main Review:**

The proposed method appears essentially an offline variant of the Dagger algorithm. The key difference seems that In Dagger, queries to the expert follows the state distribution of the student policy. In contrast, the proposed method augments a spherical (e.g. standard Gaussian) region for all expert states, and query the expert for the augmented states. In some scenarios, not needing environment interaction is an advantage.

However, the proposed method has very limited algorithmic novelty and the empirical results are not surprising. In most tasks, covariate shift begins from states similar to the demonstrations. By knowing how the expert recovers from these states, behavioral cloning has sufficient info for the policy transfer to the student.

When the proposed method is used for kick-starting, environment interaction is needed again to outperform sub-optimal experts. In this case, Dagger becomes more attractive than the proposed method, with its theoretical guarantee while being similarly easy to implement.

The empirical evaluation is also very limited, with experiments from only two environments. It is thus unclear if the method applies to other environments, In addition, the number of demonstrations may not be a fair metric for measuring the sample complexity, since the expert is queried many more times after the initial demonstration. It would be interesting to compare Dagger to the proposed method on the sample efficiency (including the additional queries to the expert).


**Time Spent Reviewing:**

4

---

> ### Author Response · Authors · 2021-08-10
> **Response to KLas**
>
> **We thank KLas for reviewing our work. Please find our response which addresses your concerns.**
>
> >*The proposed method appears essentially an offline variant of the Dagger algorithm. The key difference seems that In Dagger, queries to the expert follows the state distribution of the student policy.*
>
> **We present APC in the context of Behavior Cloning (BC) for simplicity of presentation, which is an offline learning framework. However, our method is more general than just a technique for BC and is not an offline variant of the DAgger algorithm. As we show in the paper, see Section 5.1 for example, our method is complementary to approaches like DAgger and could be combined together to get an approach which is more efficient than the original DAgger (see Figure 3). Our APC method on DAgger is literally DAgger with additional state perturbations and additional generation of corresponding actions from an expert, which are added during DAgger training iteration.**
>
> >*However, the proposed method has very limited algorithmic novelty and the empirical results are not surprising. In most tasks, covariate shift begins from states similar to the demonstrations. By knowing how the expert recovers from these states, behavioral cloning has sufficient info for the policy transfer to the student.*
>
> **The proposed method may seem to you obvious in hindsight. However, we assert that it is a quite different approach than what is typically used for data augmentation. As described in the paper, the typical approach for data augmentation in supervised learning consists in perturbing inputs but producing the same output, thus encouraging the robustness of the model. In RL, such techniques are used in [1], including Gaussian state perturbation, which in our paper is referred to as Naive ABC. Our method, however, is different, because on top of perturbing inputs, we also produce different outputs (novel actions from experts given perturbed state), encouraging the model to be feedback-responsive rather than invariant. While we point to some approaches which perform related techniques [2-4], we are unaware of any papers which systematically present an approach such as ours, which raises a question about whether this method is that obvious. One of your concerns could be that this is an obvious solution to a problem that is not frequently encountered.  However, we do not believe that this setting is actually uncommon. As we presented in the paper, the ability of querying an expert policy with continuous states may rise in different Behavior Cloning situations as well as in such widely used approaches as DAgger and kickstarting. In addition, what is more striking about the method is how much benefit it brings - see for example, Figure 1 on how much more data efficient BC becomes or Figure 3 on how it improves DAgger algorithm. Even if it may be expected that such a technique would help, we demonstrate that it actually helps quite a lot - which is solid evidence of why one wants to use it. If it was the case that it just marginally improved performance, then even if the technique may have seemed obvious, it would have been hard to argue for its usage.**
>
> >*When the proposed method is used for kick-starting, environment interaction is needed again to outperform sub-optimal experts. In this case, Dagger becomes more attractive than the proposed method, with its theoretical guarantee while being similarly easy to implement.*
>
> **Our method is really a complementary approach and as we demonstrate in the paper, it is compatible with DAgger. In fact, in Figure 3 we present results of applying the technique on top of DAgger. As described in the text, BC in this figure actually refers to DAgger with BC as the supervised learning step within DAgger and APC refers to DAgger with our method used as the supervised learning step. As you can see from the plots, using DAgger together with APC leads to much better results.**
>
> >*The empirical evaluation is also very limited, with experiments from only two environments. It is thus unclear if the method applies to other environments,*
>
> **In fact, before presenting the results, we tried our method on a few other simpler DM control suite environments - Humanoid Walk, walker Walk and Run tasks, and we saw similar results. We decided to focus our presentation on Humanoid Run and Humanoid Walls tasks because these lie on a hard part of the spectrum of the DM control suite and are generally quite challenging to solve. Results on other simpler tasks were very similar and did not add much value, so we decided not to include them. For the camera-ready version, we will add results on more tasks.**
>
> >*In addition, the number of demonstrations may not be a fair metric for measuring the sample complexity, since the expert is queried many more times after the initial demonstration.*
>
> **There are various different metrics that may be relevant, depending on the use case.  Real rollouts of either the teacher or student can be costly and this is what we keep to a minimum.  Separately, expert queries may or may not be costly, depending on the setting.  We assume real rollouts are costly and expert queries are not, which is a reasonable assumption for many, but not all settings.**
>
> References:
>
> [1] Reinforcement learning with augmented data, Laskin et al., 2020
>
> [2] Neural probabilistic motor primitives for humanoid control, Merel J. et al., 2019
>
> [3] Combining the benefits of function approximation and trajectory optimization, Mordatch I. et al., 2014
>
> [4] Interactive control of diverse complex characters with neural networks, Mordatch I. et al., 2015

---

### Official Review · Reviewer_7FbC · 2021-07-15

**Rating:** 4
**Confidence:** 4

**Summary:**

This paper proposes Augmented Policy Cloning (APC) approach to improve the data efficiency of expert behavior by augmenting expert trajectory data with virtual, perturbed states and the expert actions in these virtual states. This method is simple by applying the existing image-based data augmentation method, but it can increase the data efficiency of policy cloning and transfer high-DoF behaviors. The proposed method outperforms BC and Naive ABC.

**Limitations And Societal Impact:**

Here are a few concerns and questions I have regarding the submission, and addressing these would help me in giving a final rating:

1. I wonder why this method is limited to the high-DoF control problem. As far as I understand, this method seems to be easily applicable to other environments, but it seems that it has not been tested in diverse environments. If the experiment had been performed on other problems for which the effect was easy to show intuitively, it would have been possible to effectively show the effect and potential for general application of this method.

2. Looking at the results of Figure 2, it can be seen that the standard deviation of the results of Walls is significantly larger than that of Run. I would like to know why there is such a difference and whether there is no problem with reliability in these results.

3. Looking at Figure 4 in Section 5.2, the results of Run and Walls are slightly different. In the case of Run, there is a difference on the results of High and Medium compared to Naive ABC and BC, but it is almost the same in Low. Conversely, in the case of Walls, in the case of High and Medium, there is little difference from Naive ABC, but it is relatively more different in Low. I wonder what is the basis for this difference.

4.  I know it's due to the limitations of the amount, but putting too many details and comparisons and analyses of experiments into the supplementary material make it less legible.


*** After reading all the feedback, it seems that this paper needs more detailed explanation of all the insights and environment settings.
You should conduct more convincing experiments on the proposed method and supplement your logic.
This was reflected in my score.


**Main Review:**

This paper describes the flow of APC well and makes it easy to follow. This paper proposes APC which can generate a new state associated with the original action, and authors intuitively explain well the necessity and motivation for this method by comparing to the image-based methods. I would say the idea is not entirely new, but it can be useful in this field and the authors are well aware of its social impact.

In the significance aspect, I want to see authors to apply this method on other environment tasks. They try to propose a general scheme for more efficient transfer of expert behavior by augmenting expert's trajectory data, but their experiments seem limited in proving it. I have some questions about the experimental results and analysis (see below limitations and societal impact section).

**Time Spent Reviewing:**

Three days

---

> ### Author Response · Authors · 2021-08-10
> **Response to 7FbC**
>
> **We thank 7FbC for taking time to review our work. Below we address your concerns.**
>
> >*In the significance aspect, I want to see authors to apply this method on other environment tasks. They try to propose a general scheme for more efficient transfer of expert behavior by augmenting expert's trajectory data, but their experiments seem limited in proving it. I have some questions about the experimental results and analysis (see below limitations and societal impact section).*
>
> **In fact, before presenting the results, we tried our method on a few other simpler DM control suite environments - Humanoid Walk, walker Walk and Run tasks, and we saw similar results. We decided to focus our presentation on Humanoid Run and Humanoid Walls tasks because these lie on a hard part of the spectrum of the DM control suite and are generally quite challenging to solve. Results on other simpler tasks were very similar and did not add much value, so we decided not to include them. For the camera-ready version, we will add results on more tasks.**
>
> >*I wonder why this method is limited to the high-DoF control problem. As far as I understand, this method seems to be easily applicable to other environments, but it seems that it has not been tested in diverse environments. If the experiment had been performed on other problems for which the effect was easy to show intuitively, it would have been possible to effectively show the effect and potential for general application of this method.*
>
> **The method is not limited to the high-DoF control problems. The motivation for presenting high DoF control problems was to show the power of the method, but the method also works in lower DoF scenarios. See answer above. For camera ready versions, we would add the results on these.**
>
> >*Looking at the results of Figure 2, it can be seen that the standard deviation of the results of Walls is significantly larger than that of Run. I would like to know why there is such a difference and whether there is no problem with reliability in these results.*
>
> **The Walls task is intrinsically more diverse than Run. Each environment instantiation would correspond to a different initial position of a humanoid body together with a different setting for the walls. The wall positions make the task more or less difficult. Therefore, it is expected to see more variation in results in Walls compared to Run. In the Run task, the only variation comes from the initial position of the humanoid. We will add more clarification on this in the text.**
>
> >*Looking at Figure 4 in Section 5.2, the results of Run and Walls are slightly different. In the case of Run, there is a difference on the results of High and Medium compared to Naive ABC and BC, but it is almost the same in Low. Conversely, in the case of Walls, in the case of High and Medium, there is little difference from Naive ABC, but it is relatively more different in Low. I wonder what is the basis for this difference.*
>
> **One take away from Figure 4 is that both APC and naive ABC seem to be beneficial to kickstarting compared to BC. One of the reasons is that in the kickstarting experiment we operate in the high data regime (we have 64 acting policies and no limitation on acting/learning ratio) together with a very rich reward signal (both tasks are dense reward). In such a regime, we speculate that naive ABC may provide additional regularization for the learning and make the task easier.  In addition, it seems that for the proposed tasks, there is less variation across methods in Run, because all methods work equally well. Walls are a much harder task which could explain why methods like APC and naive ABC are so much better than BC. We will add more clarification in the main text.**
>
> >*I know it's due to the limitations of the amount, but putting too many details and comparisons and analyses of experiments into the supplementary material make it less legible.*
>
> **Thank you, we will take it into account when submitting the camera-ready version of the paper.**

---

### Official Review · Reviewer_rKs4 · 2021-07-17

**Rating:** 5
**Confidence:** 3

**Summary:**

The paper explores a novel approach to using expert policies in RL context. The main idea is to enrich expert demonstrations by perturbing a state by Gaussian noise and querying expert policy to produce an action from the perturbed state. These additional state-action pairs are used in the policy learning algorithm. The authors show experimentally that the proposed Augmented Policy Cloning (APC) can significantly improve the quality of cloned policy, compared to Behaviour Cloning.

**Limitations And Societal Impact:**

yes

**Main Review:**

The paper proposes using expert policies to perform policy cloning in RL context. The method, termed Augmented Policy Cloning, can be applied in situations where expert policy is available. The states from expert demonstrations are perturbed with Gaussian noise and expert policy is used to produce a target action. The student policy is adjusted to maximize probability of these additional state-action pairs.
The method allows to expand expert demonstrations without doing environment rollouts. Authors compare the performance of APC on two tasks(Humanoid Run and Walls) against two baselines: Behaviour Cloning and Naive Augmented Behaviour Cloning. The later(NABC) is a modification of BC  via state perturbations but without changing corresponding actions.In the regime of low number of expert demonstrations APC significantly outperforms BC and NABC.

This idea of using expert policies for policy cloning is new to me. It looks like a useful tool in practice for efficiently using expert policies without the need to perform rollouts.

In terms of presentation quality I found the paper hard to read. Especially experimental sections can be improved by highlighting high level results and potentially moving some details into the Appendix.
The detailed comparison of APC and BC/NABC in Fig. 1 seems not that informative to me: it is clear that using expert queries around demonstration state will give much more information to the policy then in BC case. I think to make experimental part more informative it would be very valuable to consider a set of standard continuous-control tasks (like form DM control suite) and describe appropriate levels of state perturbation and advantage in reward.

I also found the Naive ABC not a very meaningful baseline to compare to: simply perturbing the state with Gaussian noise will unlikely lead to a state where the same action is optimal. From Fig 1, 2, 3 it does look that NABC performs slightly worse then BC. However, in the case of kickstarting interestingly Naive ABC performs on par with proposed APC. I would be interested in authors thoughts on why this is the case.

**Time Spent Reviewing:**

4

---

> ### Author Response · Authors · 2021-08-10
> **Response to rKs4**
>
> **We thank rKs4 for taking time to write a review for our work. Below we address your concerns.**
>
> >*In terms of presentation quality I found the paper hard to read. Especially experimental sections can be improved by highlighting high level results and potentially moving some details into the Appendix*
>
> **Thank you for highlighting this. We would focus on improving the clarity of the paper and especially of the experimental section in the camera-ready version.**
>
> >*The detailed comparison of APC and BC/NABC in Fig. 1 seems not that informative to me: it is clear that using expert queries around demonstration state will give much more information to the policy then in BC case.*
>
> **In Figure 1 and 2, we wanted to highlight that APC performs more efficiently than a standard technique of learning from expert data - BC and a naive technique of augmenting states - naive ABC. On one hand, it is intuitively expected that using additional virtual data via expert queries should help, but Figures 1 and 2 demonstrate the order of improvement, i.e., for Run task, 10 trajectories may be enough to learn close to optimal policy, whereas it takes BC around 500 to learn similar performance. We believe that this is not entirely expected.  Moreover, we are not aware of an approach similar to ours having been previously proposed.**
>
> >*I think to make experimental part more informative it would be very valuable to consider a set of standard continuous-control tasks (like form DM control suite) and describe appropriate levels of state perturbation and advantage in reward.*
>
> **In fact, before presenting the results, we tried our method on a few other simpler DM control suite environments - Humanoid Walk, Walker Walk and Run tasks, and we saw similar results. We decided to focus our presentation on Humanoid Run and Humanoid Walls tasks because these lie on a hard part of the spectrum of the DM control suite and are generally quite challenging to solve. Results on other simpler tasks were very similar and did not add much value, so we decided not to include them. For the camera-ready version, we will add results on more tasks.**
>
> >*I also found the Naive ABC not a very meaningful baseline to compare to: simply perturbing the state with Gaussian noise will unlikely lead to a state where the same action is optimal. From Fig 1, 2, 3 it does look that NABC performs slightly worse then BC. *
>
> **Basic augmentation usually involves perturbing inputs in the hope of making the model robust / insensitive to these perturbations. While we do not believe this is well motivated in the control setting, it does constitute the naive approach for data augmentation, if simply adapted from other settings (e.g. supervised learning).  Beyond this, we also felt that Naive ABC is a reasonable scientific “control” insofar as we did not expect it to necessarily work better than BC, but its inclusion gives us a fairer comparison.**
>
> >*However, in the case of kickstarting interestingly Naive ABC performs on par with proposed APC. I would be interested in authors thoughts on why this is the case.*
>
> **The kickstarting experiment is quite different from the two others, as it is a high data regime (we have a distributed training with 64 acting policies without any control of relative learning/acting ration) and there is a strong reward signal present (we have dense reward tasks). We think that under such conditions, the naive ABC adds additional regularization to the training. Moreover, as could be seen in Figure 1 and 2, APC makes a big difference in low-data regime. In a high-data regime of kickstarting together with a strong reward signal, the advantage is expected to be smaller.**

---

### Author Response · Authors · 2021-08-10
**Thank you for the reviews**

We would like to thank all the reviewers for having taken the time to review our work.

We hope that our answers clarify mis-understanding and address your concerns.

Thank you.

---

### Decision · Program_Chairs · 2021-09-28

**Decision:**

Reject

**Comment:**

The ideas presented in the paper have been deemed interesting by most reviewers. However a few issues have been identified:

- The experimental setting is limited and authors should motivate better their choice of environments.

- The motivation for the proposed strategy needs to be discussed further. At the moment the use case setting for the proposed method appears quite limited.

**Consistency Experiment:**

NeurIPS has a long history of experimentation. In 2014, NeurIPS ran an experiment in which 10% of submissions were reviewed by two independent committees to quantify the randomness in the review process. This year, we repeated a variant of this experiment to see how the quality of the review process has changed over time.  This paper was part of the experiment and was therefore assigned to two committees (consisting of reviewers, an Area Chair, and a Senior Area Chair) that reached independent decisions.  If both committees made the same recommendation, this recommendation was followed. If a single committee recommended acceptance, the paper was accepted (with the exception of a few cases in which the other committee identified what we considered a fatal flaw, e.g., an error in a key result).

Both committees reached the same decision: **Reject**

The other committee assigned to the paper recommended **Reject**.  You can find the other set of reviews, along with any follow up discussion with the authors here:
https://openreview.net/forum?id=rOkwuRct3kvDm